# Processability and Mechanical Properties of Thermoplastic Polylactide/Polyhydroxybutyrate (PLA/PHB) Bioblends

**DOI:** 10.3390/ma14040898

**Published:** 2021-02-14

**Authors:** Olga Olejnik, Anna Masek, Jakub Zawadziłło

**Affiliations:** 1Institute of Polymer and Dye Technology, Faculty of Chemistry, Lodz University of Technology, ul. Stefanowskiego 12/16, 90-924 Lodz, Poland; olga.olejnik@dokt.p.lodz.pl; 2Faculty of Chemistry, Wroclaw University of Science and Technology, Wybrzeże Wyspianskiego 27, 50-370 Wroclaw, Poland; jakub.zawadzillo@pwr.edu.pl; 3Institute of Polymer Materials, Leibniz Institute of Polymer Research Dresden, Hohe Str. 6, D-01069 Dresden, Germany

**Keywords:** polylactide, polyhydroxybutyrate, blend, bioplastic

## Abstract

This work considers the application of eco-friendly, biodegradable materials based on polylactide (PLA) and polyhydroxybutyrate (PHB), instead of conventional polymeric materials, in order to prevent further environmental endangerment by accumulation of synthetic petro-materials. This new approach to the topic is focused on analyzing the processing properties of blends without incorporating any additives that could have a harmful impact on human organisms, including the endocrine system. The main aim of the research was to find the best PLA/PHB ratio to obtain materials with desirable mechanical, processing and application properties. Therefore, two-component polymer blends were prepared by mixing different mass ratios of PLA and PHB (100/0, 50/10, 50/20, 40/30, 50/50, 30/40, 20/50, 10/50 and 0/100 mass ratio) using an extrusion process. The prepared blends were analyzed in terms of thermal and mechanical properties as well as miscibility and surface characteristics. Taking into account the test results, the PLA/PHB blend with a 50/10 ratio turned out to be most suitable in terms of mechanical and processing properties. This blend has the potential to become a bio-based and simultaneously biodegradable material safe for human health dedicated for the packaging industry.

## 1. Introduction

Biopolyesters, including polyhydroxybutyrate and polylactide, are of great interest to researchers [1,2,3,4,5]. These polymers are polyesters that are synthesized using bio-resources such as sugar or plant oil. On the one hand, the whole polymerization process can be conducted via enzymatic reactions giving rise to the polymers. On the other hand, biosynthesis may lead to the creation of substrates, which are used in a further, non-biological process to synthesize biopolyester [6,7]. Biopolyesters are biodegradable and biocompatible, which makes them very attractive from the point of view of the packaging industry [8] as well as the medical industry [9]. They can be used as packaging material [10] or as a drug delivery system [11]. In addition, biopolyesters such as polylactide or polyhydroxybutyrate can be successfully used in 3D printing technology [12].

As far as the packaging and medical industries are concerned, the processing and modification of biopolyesters should be carried out in a way that does not alter their biodegradability and biocompatibility [13]. Various modifiers, including fillers, antioxidants, and processing aids, can be harmful to the environment, which makes the resultant “bio” composite dangerous as well. This is why the selection of additives that are not harmful to the environment is of great interest to scientists and engineers [14]. Biopolyesters can also be modified via polymer–polymer blending. This route allows the creation of composites with unique properties. Moreover, this technique is much easier, faster and cheaper than other polymeric material creation methods, such as copolymerization [15]. Polylactide can be blended with both petropolymers [16,17] and biopolymers [18,19,20]. Unfortunately, it is very often essential to use polymer blend compatibilizers due to chemical and structural differences as well as incompatibility related to the surface energy or viscosity of components [21]. It is also noteworthy that, in many cases, the miscibility of selected polymers, including biopolyesters, may depend on their molecular mass [22]. Moreover, blends based on biopolymers are susceptible to hydrolysis, which occurs especially during high-temperature processing in the presence of moisture [23]. Because of this phenomenon, it is essential to achieve process repeatability, for example by processing window broadening. Nevertheless, blending polylactide with other biopolyesters leads to the creation of composite materials that are environmentally friendly [24]. Thus, it is still necessary to overcome problems and create novel, pro-ecological materials with desirable properties.

Both polylactide (PLA) and polyhydroxybutyrate (PHB) are bio-based and biodegradable polyesters and are characterized by good biocompatibility, biodegradability and sustainability. These polymers can be compared with conventionally used plastics in terms of the thermal and mechanical properties [22]. Therefore, polyhydroxybutyrate and polylactide can be used as transparent films in the food packaging industry because of their biodegradability and biocompatibility, which are extremely necessary nowadays [23]. PLA and PHB are also suitable for the production of disposable cutlery [24] and biomedical implants [25]. Separately, PLA and PHB are characterized by poor processing [26], but such properties of their blends have not been interpreted. Moreover, these polymers are brittle at room temperature. Nevertheless, blending polylactide with polyhydroxybutyrate, which is a highly crystalline biopolyester, results in new materials with interesting physical, thermal and mechanical properties in comparison to neat polylactide (PLA) [27]. Blending polylactide with polyhydroxybutyrate is not a new idea [28,29]. On the basis of information available in the literature, mixing PLA with 25 wt% of PHB provided a reinforcement effect and better mechanical properties in comparison to neat PLA, which was observed by Zhang et al. [15]. Furthermore, some interactions between both polymers were also noticed. Crystallization behavior of poly(L-lactic acid) affected by the addition of a small amount of poly(3-hydroxybutyrate) was researched by Hu et al. [30]. The impact of different molecular weight PHB on the PLA matrix was also detected. Nevertheless, most of the research concerns poly(3-hydroxybutyrate), and there is lack of information about commercially available PHB containing 12 mol% 4-hydroxybutyrate, which could be of interest to both scientists and producers. 

Most of the research concerning PLA/PHB blends is focused on testing films, and there is no information about thicker forms of specimens that could be dedicated to stiffer packaging, including boxes. The processability of such blends is also insufficiently described. Furthermore, the best mass ratio of PLA and PHB to obtain a blend with the most satisfactory properties has not been reported. Thus, we decided to create PLA/PHB blends from commercially available pellets in order to obtain materials with desirable mechanical, processing and application properties that can be dedicated to environmentally friendly packaging. 

## 2. Materials and Methods 

### 2.1. Reagents

The first component of the blend was polylactide (PLA), IngeoTM 2003D (provided by NatureWorks LLC, Minnetonka, MN, USA) with a glass transition temperature of 328–333 K, peak melt temperature of 418–433 K and with a melt flow rate of 6 g/10 min (conditions: temperature 483 K, nominal load 2.16 kg). The density of this material amounts to 1.24 g/cm^3^. The second ingredient was polyhydroxybutyrate (PHB), which belongs to the polyhydroxyalkanoate group of polymers (PHA) and was obtained from Simag Holdings LTD (Hong Kong, China). This material, containing 12 mol% 4-hydroxybutyrate, is characterized by an average molecular weight of 520 kDa and a density of 1.25 g/cm^3^. The melt flow rate is 18 (conditions: temperature 443 K, nominal load 2.16 kg), and the moisture content is 0.05%. 

### 2.2. Samples Preparation

The selected PLA and PHB pellets were first dried at 343 K in a vacuum oven for 24 h. Then, PLA/PHB mixtures were prepared by manually pre-mixing selected pellets in a laboratory beaker according to the composition presented in Table 1. The prepared mixture was processed using a single-screw laboratory extruder (Zamak Mercator, Skawina, Poland) equipped with a 25 mm screw diameter and characterized by a L/D ratio of 24. The temperature of the extrusion process amounted to 453 K, the screw rotation speed was about 40 rpm, and pressure equaled 17 atm. The flat tape of blends was formed using a special flat slit extrusion head. The estimated output of the process amounted to 8 kg/h. The formed flat tape was cooled in the air at room temperature. Finally, the tape was cut into smaller samples, which were 150 mm long, 25 mm wide and 1 mm thick.

### 2.3. Measurement of Mixing Energy

The investigation of mixing energy and torque was carried out in a Brabender micromixer in conjunction with the WinMix program (version 3.0.0) [31]. The experiment consisted in filling the Brabender mixing chamber (Brabender GmbH & Co. KG, Duisburg, Germany) with a 50 g mixture of pure polylactide (PLA) and polyhydroxybutyrate (PHB) granulate in selected ratios. The temperature of the micromixer was about 453 K. After filling the mixer, torque as a function of time was recorded by the WinMix program. The mixing energy was calculated from the resistance shown by the mixed granules. This investigation not only measured mixing energy but also determined the role of each component, including plasticizing properties, by comparing torque and mixing energy at the selected temperature.

### 2.4. Differential Scanning Calorimetry

Differential Scanning Calorimetry (DSC) was utilized to determine the glass transition temperature, the temperature of cold crystallization and the melting point of the polymeric blends. The research was performed with a DSC1 device provided by Mettler Toledo (TA 2920, TA Instruments, Greifensee, Switzerland). One of the aluminum pans was filled with about 5–7 mg of the sample, and the second one acted as a reference. The analysis was conducted using the following parameters: the first cycle (heating from 273 K to 473 K) was performed under a dynamic flow of argon at a rate of 50 mL/min for 15 min, and the second cycle was performed at 473 K under a dynamic flow of argon at a rate of 50 mL/min for 10 min. The third cycle (cooling from 473 K to 273 K) was performed under a dynamic flow of argon at a rate of 50 mL/min for 15 min, and the fourth cycle (heating from 273 K to 623 K) was performed under a dynamic flow of air at a rate of 50 mL/min. The heating rate amounted to 20 K/min. The temperatures of the phase transitions of the samples and their specific heat were determined for the different compositions on the basis of the DSC curves.

### 2.5. Melt Flow Rate Determination

The melt flow rate is one of the most important parameters in terms of polymer processing, including injection molding, extrusion, thermoforming or printing. The measurement was conducted according to ISO 1133D with the use of a MeltFloWon plus melt index tester (CEAST, Planegg, Germany). The measurement was performed by utilizing typical parameters used for polylactide research, including a temperature of 463 K and a mass of 2.16 kg.

### 2.6. Mechanical Properties

The measurements were carried out using a Zwick 1435 test machine (DEGUMA-SCHÜTZ GmbH, Geisa, Germany) in accordance with PN-ISO 527-1/-2, using flat strip test pieces with a width of 25 mm and a length of 80 mm. The parameters of maximum tensile strength (T_Fmax_), tensile strength at break (TS), elongation at break (Eb) and elongation at maximum strength (E_Fmax_) were obtained. The measurements were carried out at a rate of 50 mm/min and initial force of 0.1 N.

### 2.7. Dynamic Mechanical Analysis

Dynamic mechanical analysis is one of the most useful techniques for characterizing materials to obtain parameters that are relevant from the processing and end-use points of view. An Ares G2 rheometer provided by TA Instruments (New Castle, DE, USA) was used. The measurements were carried out as a function of rotor rotation frequency from 0 to 625 rad/s at a constant temperature of 453 K, and the storage and loss moduli, viscosity and damping characteristics were determined. The test pieces were discs with a diameter of 20 mm and thickness of 1 mm.

### 2.8. Surface Characterization of Polymer Blends

The surface character is crucial in terms of the resistance of biopolymers to different solvents and their compatibility with other materials or living organisms. Information about polar and dispersive components of surface energy can be obtained from suitable calculations. To calculate surface energy and its components, the Owens–Wendt–Rabel–Kaelble (OWRK) method was used. In this technique, the contact angle was calculated using a goniometer model OCA15EC (DataPhysics Instruments GmbH, Filderstadt, Germany) and three liquids with different polarity: water, diiodomethane and ethylene glycol. The volume of a drop was about 1.2 µL, and the speed of dosing was 2 µL/s. The surface energy was calculated using the Young equation: *γS* = *γSL* + *γL**cos*θ(1)
where *γS* is the surface energy of solid phase, *γSL* is the surface energy at the solid–liquid interphase and *γL* is the surface energy of liquid phase.

## 3. Results

### 3.1. Measurement of Mixing Energy

According to the results presented in Figure 1, the mixing energy of pure PLA amounted to 0.69 kNm/g and diminished in proportion to the polyhydroxybutyrate (PHB) content (%) in a blend. As can be seen, pure polylactide is stiffer and shows greater resistance to mixing than polyhydroxybutyrate (PHB). The mixing energy of pure PLA is about 6 times higher than that of pure PHB. Polyhydroxybutyrate (PHB) makes polylactide PLA easier to process, which was observable by the lower mixing energy after adding the PHB to the PLA. The higher the amount of PHB, the lower the mixing energy. 

### 3.2. Differential Scanning Calorimetry (DSC) Analysis

Differential scanning calorimetry provides information about temperatures of glass transition, cold crystallization, and melting point. Moreover, this measurement can be useful for determining the compatibility of polymer blends. It can be observed that the phase transition temperature of composites depends on the intermolecular interactions between the chains of these polymers. In the case of compatible materials, the interaction forces were stronger, and a single temperature for a specific phase transition was observed. If a composite consisted of incompatible components, two temperatures for glass transition, crystallization and melting point were visible in the DSC curve. The DSC curves of selected PLA/PHB blends are presented in Figure 2. According to this figure, PLA/PHB blends were characterized by two visible melting peaks. The first peak characterized one type of crystal melting, probably corresponding to PLA, while the second one possibly related to PHB crystal melting, which was also detected by Zhang et al. [15]. Nevertheless, the second peak of every blend was responsible for whole material melting, and this parameter is noted in Table 2 as melting point (T_m_). 

According to Figure 2 and Table 2, blends with an equal or higher polyhydroxybutyrate (PHB) content had lower glass transition (T_g_) temperatures than materials with a higher amount of polylactide (PLA). This parameter decreased for these blends in a linear way from 333 K to 325 K as a function of polyhydroxybutyrate (PHB) content. The addition of smaller amounts of polyhydroxybutyrate (PHB) to polylactide (PLA) caused an irregular drop in this parameter. Nevertheless, it can be assumed that polyhydroxybutyrate (PHB) acts as plasticizer, in relation to polylactide, because every blend containing this biopolyester had lower T_g_ in comparison to pure polylactide. Furthermore, only one glass transition was observed in the DSC.

According to Figure 2 and Table 2, the temperature of cold crystallization (T_cc_), similarly to glass transition temperature (T_g_), decreased with the increase in polyhydroxybutyrate content. This parameter changed in a linear way. The blend containing the lowest amount of polyhydroxybutyrate (PHB) had the highest value of about 394 K, which was lower than that of pure PLA by about 6 degrees. The blend with the highest PHB content had the lowest cold crystallization temperature of about 366 K, which was higher than the value for pure PHB by about 13 degrees. This means that the higher the PHB content, the faster the blend crystalizes at lower temperatures. The enthalpies of blend phase changes also show that the addition of PHB led to an increase in the crystal phase, which was also observed by Hu et al. [30].

Blending a small amount of polyhydroxybutyrate (PHB) with polylactide (PLA) caused a decrease in melting point from 429 K to 426 K in comparison to pure PLA, which can be seen in Figure 2 and Table 2. Polymer blends of 43% or higher PHB content revealed a higher melting point than pure polylactide. Only a small addition of polyhydroxybutyrate can be useful for reducing the melting point, which corresponds to the material processing temperature and acts as a plasticizer.

### 3.3. Melt Flow Rate Determination

Measurement of melt flow rate also confirms the plasticizing effect of polyhydroxybutyrate addition (Figure 3a,b). The relationship between melt volume rate (MVR) and the amount of polyhydroxybutyrate was exponential. It ranged from 10 to 302 cm^3^/10 min, and higher values of MVR were related to the polymer blends with greater amounts of polyhydroxybutyrate. Viscosity of the blends also depends on the polyhydroxybutyrate content, and the higher the PHB content, the lower the viscosity. This dependence also took an exponential form.

### 3.4. Mechanical Properties

According to Figure 4, the addition of a small amount of PHB resulted in the improvement in mechanical properties. Similar observations were noted by Zhang et al. [15], wherein the enhancement was explained as a fine dispersion of PHB crystals in the PLA matrix such that the crystals acted as a filler. Too many PHB crystals cannot be well dispersed, and this causes deterioration in mechanical properties. The maximum tensile strength (T_Fmax_) in prepared blends depended on their PLA/PHB composition and significantly diminished as a function of polyhydroxybutyrate content, but it should be noted that the correlation was not perfectly linear (Figure 5). The highest value was about 47 MPa for the PLA/PHB 50/10 blend, and the lowest was 26 MPa for PLA/PHB 10/50. The elongation at maximum strength (E_Fmax_) does not depend significantly on blend composition, and the values of this parameter ranged from 2.68% to 3.84%. There was a slight linear decrease in this parameter as a function of PHB content.

The dependence of polyhydroxybutyrate content on tensile strength at break (TS) looked similar to maximum tensile strength (T_Fmax_). According to Figure 5, the higher the PHB content, the lower the TS value. The correlation between the 30/40, 50/50 and 40/30 blends was the most interesting because a decrease in this parameter from 32 MPa to 7 MPa was observed. The elongation at break also depends on PLA/PHB composition and rises in an exponential way as a function of PHB content.

### 3.5. Dynamic Mechanical Analysis (DMA)

The values of storage and loss moduli changed in an irregular way (Figure 6). The behavior and dynamic properties of material based on PLA with PHB cannot be predicted by the content of a component in the blend. The PLA/PHB 50/20 blend was the only material that revealed an evident intersection of G’ and G” in the range of the investigated angular frequency. In contrast to the other blends, blends of 50/10, 40/30 and 30/40 PLA/PHB were close to having an intersection in the definite range of angular frequency. The lowest value of loss angle was for the PLA/PHB 50/20 blend, which contained about 28% of polyhydroxybutyrate (PHB) (Figure 6b). A higher value of tgδ was found for the PLA/PHB 30/40 blend containing about 57% of PHB, but the lower PHB content of about 43% resulted in higher values of G”/G’. The dynamic viscosity results for polymer blends differed from the static ones. The highest values of dynamic viscosity were obtained for the PLA/PHB 50/20 blend and the lowest ones for the PLA/PHB 10/50 blend (Figure 7a). No closer correlation between this parameter and PHB content was found.

### 3.6. Surface Characterization of Polymer Blends

The contact angle results were useful for the calculation of surface energy (SE) and its components (Figure 8). It can be noted that there was no visible correlation between surface energy and polyhydroxybutyrate content nor the polar or dispersive component of SE. Nevertheless, almost every result showed that the specimens had a hydrophobic character. Only the 40/30 PLA/PHB blend seemed to be less hydrophobic in comparison to the materials.

## 4. Conclusions

On the basis of the measurement results, it can be noted that torque as well as mixing energy decrease with higher content of polyhydroxybutyrate (PHB) in the PLA/PHB mixture. Therefore, in such a blend, polyhydroxybutyrate facilitates the processability of the polylactide matrix. In spite of the higher melting point of polyhydroxybutyrate (PHB), which was detected by Differential Scanning Calorimetry (DSC), the addition of PHB to the blend causes an increase in melt volume flow rate, which also facilitates material processing. The addition of polyhydroxybutyrate (PHB) to polylactide causes a decrease in glass transition temperature (T_g_). The measurement of mechanical properties reveals that tensile strength at break (TS) and maximum tensile strength (T_Fmax_) diminish as a function of (PHB) content, but a small amount of PHB leads to material enhancement. On the other hand, elongation at break (E_b_) rises with the polyhydroxybutyrate (PHB) content but is lower in comparison to neat PLA. Dynamic Mechanical Analysis (DMA) results showed that PLA/PHB blends are shear-thinning fluids, regardless of their composition. Only the 50/20 PLA/PHB blend had G”/G’ lower than 1 in the frequency of 500–625 rad/s. Moreover, the contact angle measurements provided information about the hydrophobicity of the blends. Polylactide is a stiffer material than polyhydroxybutyrate; thus, blends prepared from these two polymers have intermediate properties, including mixing energy, glass transition temperature and viscosity or melt flow rate. Polyhydroxybutyrate facilitates the processability of polylactide, so these blends could become novel materials in the packaging sector. Based on the test results, the PLA/PHB blend with a 50/10 ratio seems to be the most appropriate because of its mechanical and processing properties. These results support the idea that thermodynamic mixing and energetics of the polymer–polymer interface are critical design parameters for creating highly pro-ecological polymeric materials. The presented blend has the potential for replacing and being safer than non-biodegradable materials, which contain additives harmful to human beings. 

## Figures and Tables

**Figure 1 materials-14-00898-f001:**
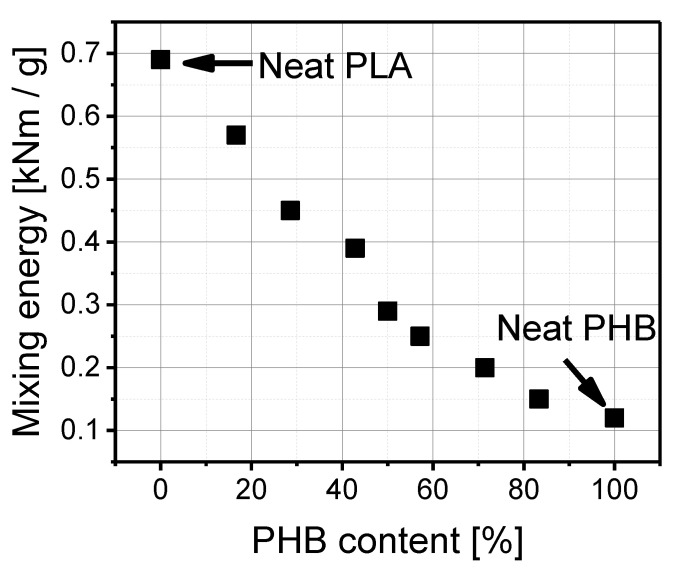
Mixing energy as a function of PHB content (%) in a blend.

**Figure 2 materials-14-00898-f002:**
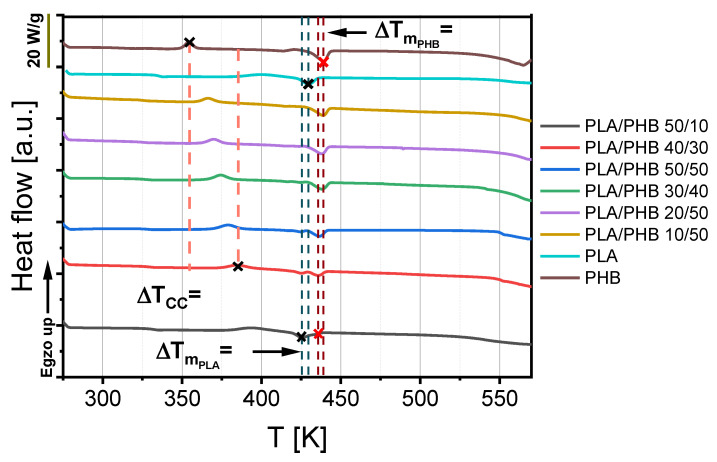
DSC curves of polylactide (PLA), polyhydroxybutyrate (PHB) and their blends.

**Figure 3 materials-14-00898-f003:**
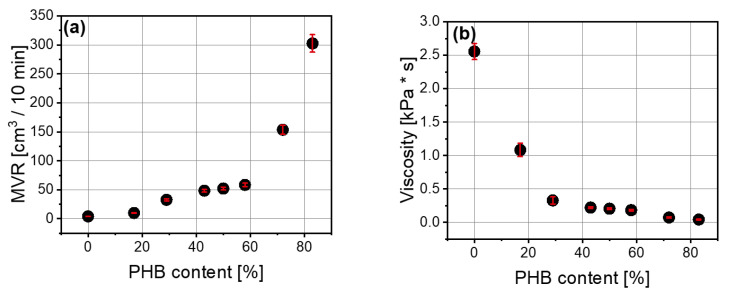
Dependence of melt volume flow rate (cm^3^/10 min) (**a**) and viscosity (**b**) on polyhydroxybutyrate content (%) in PLA/PHB blend.

**Figure 4 materials-14-00898-f004:**
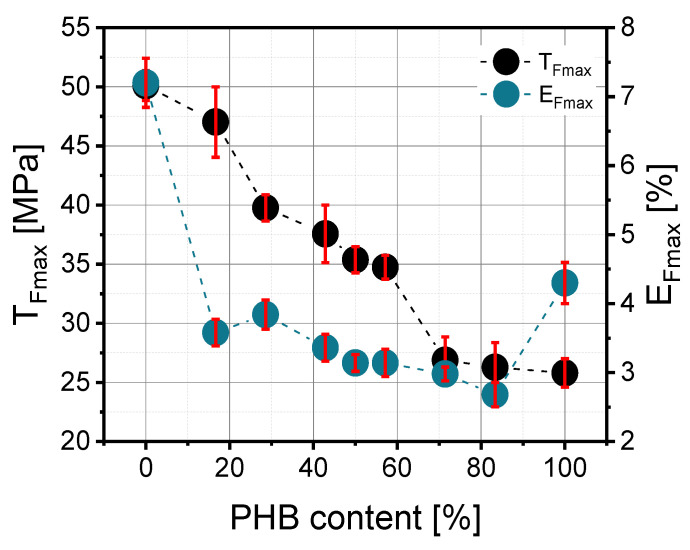
Dependence of maximum tensile strength (T_Fmax_) and elongation at ultimate strength (E_Fmax_) on PHB content in the PLA/PHB blend.

**Figure 5 materials-14-00898-f005:**
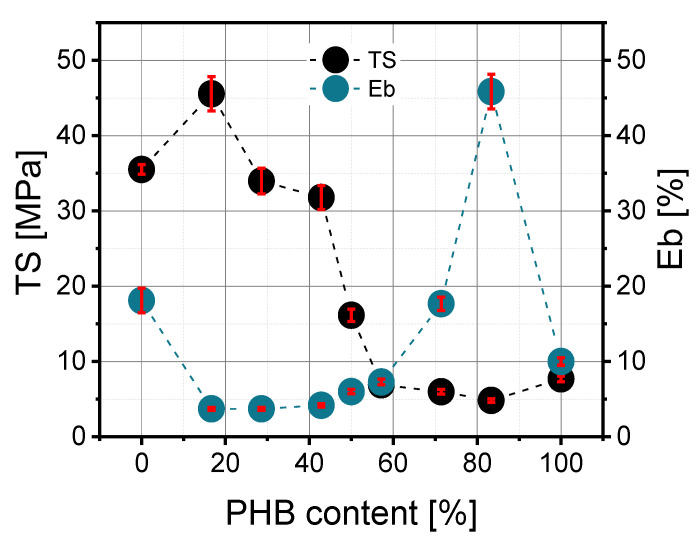
Dependence of tensile strength at break (TS) and elongation at break (Eb) on PHB content in the PLA/PHB blend.

**Figure 6 materials-14-00898-f006:**
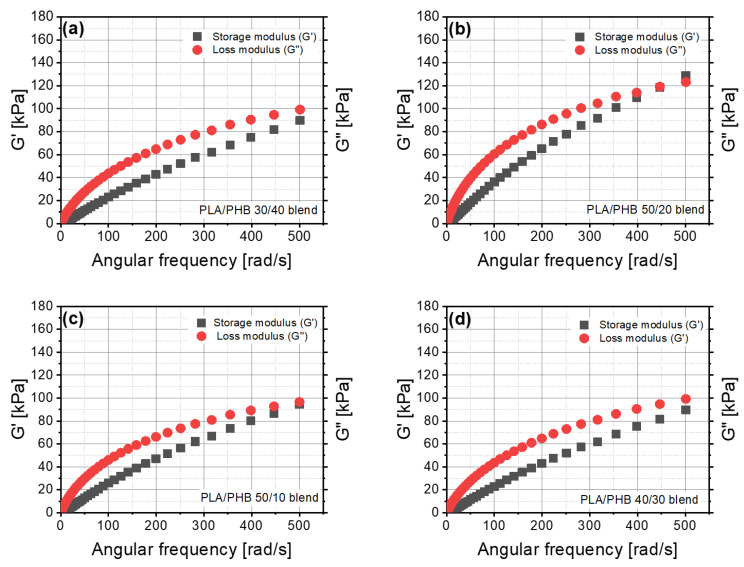
Storage and loss moduli as a function of angular frequency for the: (**a**) 30/40 PLA/PHB blend, (**b**) 50/20 PLA/PHB blend, (**c**) 50/10 PLA/PHB blend, (**d**) 40/30 PLA/PHB blend.

**Figure 7 materials-14-00898-f007:**
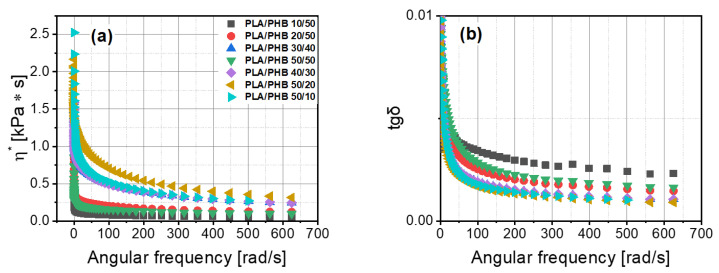
Viscosity of PLA/PHB blends (**a**) and loss tangent coefficient (tgδ) (**b**) as a function of angular frequency.

**Figure 8 materials-14-00898-f008:**
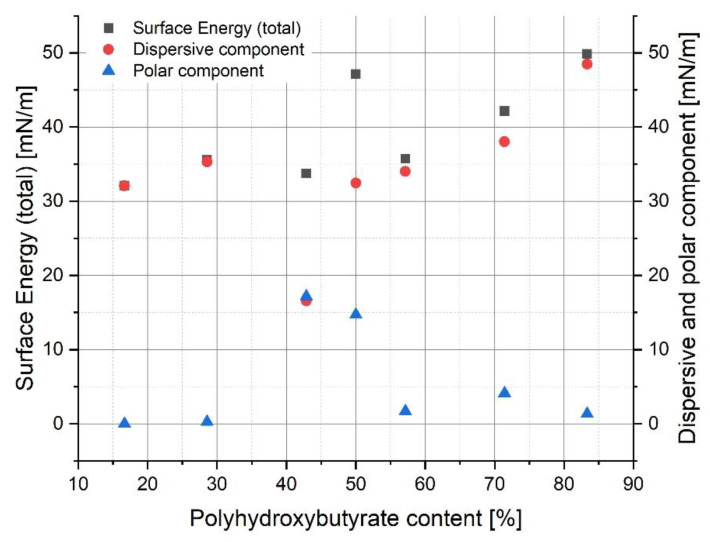
Dependence of surface energy and its components on polyhydroxybutyrate content in a polymer blend.

**Table 1 materials-14-00898-t001:** Composition of two-component polymer blends based on polylactide and polyhydroxybutyrate.

Ingredient	Polymer Blend (Mass Ratio)								
Polylactide (PLA)	100	-	50	50	40	50	30	20	10
Polyhydroxybutyrate (PHB)	-	100	10	20	30	50	40	50	50

**Table 2 materials-14-00898-t002:** Composition of two-component polymer blends based on polylactide (PLA) and polyhydroxybutyrate (PHB).

Sample	PHB Content [%]	T_g_ [K]	ΔH_cc_ [J/g]	T_cc_ [K]	ΔH_m_ [J/g]	T_m_ [K]
PLA	0	335	14.6	400	15.6	429
PHB	100	317	15.4	355	33.9	439
PLA/PHB 50/10	17	332	23.1	394	19.8	426
PLA/PHB 40/30	43	334	22.4	385	23.2	435
PLA/PHB 50/50	50	331	22.4	379	26.0	436
PLA/PHB 30/40	57	329	19.5	374	24.8	438
PLA/PHB 20/50	71	327	18.4	370	28.8	438
PLA/PHB 10/50	83	325	17.6	366	30.4	438

**NOTE:** where T_g_—glass transition, ΔH_cc_—cold crystallization enthalpy, T_cc_—temperature of cold crystallization, ΔH_m_—enthalpy of melting, T_m_—melting point.

## Data Availability

Data sharing is not applicable to this article.

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
