# Peer review of "Processability and Mechanical Properties of Thermoplastic Polylactide/Polyhydroxybutyrate (PLA/PHB) Bioblends"

_materials, 2021, doi:10.3390/ma14040898_

Round 1
Reviewer 1 Report
The paper by Olejnik et al is a study on PLA/PHA blends that aims to address the environmental issue for using as a replacement for traditional plastics. This approach is not novel as both PLA and PHA are well known. The idea of the authors would be to avoid the use of any additives which maybe dangerous for human or for the environment. However, the polymers used are both commercially sourced therefore the authors should firstly address the presence of any additives in the polymers they have used. Secondly, the use of additives is largely motivated by the applications for the blends. Just to make an example, viscosity modifiers are needed when the polymers needs to be processed in high volume processing like film blowing or similar. In such case, the research should be intended to develop natural replacements if the blends is not showing the properties needed. This point is not clearly addressed in the paper. For example, in line 167 the authors wrote “Polyhydroxybutyrate (PHB) acts like a plasticizer and the higher the amount of PHB, the lower is the mixing energy”. This statement is interesting, but it must be contextualized with some references supporting the industrial relevance as replacement for non-natural plasticizers. Similarly, as the authors have access to an ARES rheometer and to Melt Flow equipment they could perform deeper analysis to demonstrate the industrial impact and application or their blends. Looking in the Natureworks site it is clearly stated that Ingeo 2003D “is specifically designed for use in fresh food packaging and food
serviceware applications” (https://www.natureworksllc.com/~/media/Technical_Resources/Technical_Data_Sheets/TechnicalDataSheet_2003D_FFP-FSW_pdf.pdf) therefore, the advantages to mixing with PHA should be better supported.
Paragraph 2.2 The paragraph should be rewritten reporting the information about the Brabender model used for sample preparation and all the conditions (i.e. temperature, screw speed etc) should be given. The authors report that flat strip samples are produced with a thickness of 1 mm. I suppose that a flat slit extrusion head has been used. The film were then collected on a rotating spool, can the authors give more details about the spinning speed.
Usually, compounding is performed in twin screw extruders and the pellets obtained therefrom are then processed in single screw extruders. The authors directly used a single screw, how were the polymers mixed to ensure the right mixing ratio. This is usually ensured by using gravimetric feeders.
All the mechanical properties are reported with not evidence of error bars and no statistical analysis is performed to assess the p-value. This should be done to support the data analysis.
Author Response
Institute of Polymer and Dye Technology
Technical University of Lodz
90-924 Lodz, ul Stefanowskiego 12/16, Poland
Tel.: +48 42 631 32 23, Fax: +48 42 636 25 43
January 11 2020
Materials
Dear Professor,
We are resubmitting our revised paper entitled Thermoplastic and biodegradable polymer blends from plant materials by, Olga Olejnik, Anna Masek and Jakub Zawadziłło with a request to reconsider it for publication in Materials.
We have carefully considered the Editor and Reviewers' comments. The manuscript was revised exactly according to these comments. The list of responses to the reviewers’ comments and corrections made in the manuscript is attached.
The manuscript has not been previously published, is not currently submitted for review to any other journal, and will not be submitted elsewhere before a decision is made by this journal.
For correspondence please use the following information:
corresponding author: Anna Masek
Institute of Polymer and Dye Technology
Technical University of Lodz
90-924 Lodz, ul Stefanowskiego 12/16, Poland
Tel.: +48 42 631 32 93
Fax: +48 42 636 25 43
e-mail: anna.masek@p.lodz.pl
Yours sincerely,
Ph. D., D.Sc. Anna Masek
Answers to reviewer #1 comments
Reviewer #1: The paper by Olejnik et al is a study on PLA/PHA blends that aims to address the environmental issue for using as a replacement for traditional plastics. This approach is not novel as both PLA and PHA are well known. The idea of the authors would be to avoid the use of any additives which may be dangerous for human or for the environment. However, the polymers used are both commercially sourced therefore the authors should firstly address the presence of any additives in the polymers they have used.
Answer 1 for Reviewer #1: We are thankful for the Reviewer’s comment. At the moment we are not able to assess the purity of polymer pellets and we are basing on manufacturers information and own FT-IR spectra. Nevertheless our paper is focused on creating new blends by mixing commercially available biopolymer pellets which are dedicated to food packaging and shows that addition of PHB to PLA improves PLA processability. Moreover, we can obtain better mechanical properties by choosing proper mass ratio of used biodegradable and bio-based thermoplastics.
Reviewer #1: Secondly, the use of additives is largely motivated by the applications for the blends. Just to make an example, viscosity modifiers are needed when the polymers needs to be processed in high volume processing like film blowing or similar. In such case, the research should be intended to develop natural replacements if the blends is not showing the properties needed. This point is not clearly addressed in the paper. For example, in line 167 the authors wrote “Polyhydroxybutyrate (PHB) acts like a plasticizer and the higher the amount of PHB, the lower is the mixing energy”. This statement is interesting, but it must be contextualized with some references supporting the industrial relevance as replacement for non-natural plasticizers. Similarly, as the authors have access to an ARES rheometer and to Melt Flow equipment they could perform deeper analysis to demonstrate the industrial impact and application or their blends. Looking in the Natureworks site it is clearly stated that Ingeo 2003D “is specifically designed for use in fresh food packaging and food serviceware applications” (https://www.natureworksllc.com/~/media/Technical_Resources/Technical_Data_Sheets/TechnicalDataSheet_2003D_FFP-FSW_pdf.pdf) therefore, the advantages to mixing with PHA should be better supported.
Answer 2 for Reviewer #1: We thank the Reviewer for paying attention to these problems. We wanted to show, that blending PLA with the proper amount of PHB leads to obtain bio-based and biodegradable thermoplastic material with better processability and good mechanical properties. Maybe the sentence “Polyhydroxybutyrate (PHB) acts like a plasticizer and the higher the amount of PHB, the lower is the mixing energy” is not clear and it is better to write: “Polyhydroxybutyrate (PHB) makes polylactide PLA easier to process, which is observable by lower mixing energy after adding the PHB to PLA.” The measured melt flow index (MFI) also indicates, that prepared blends are more easily processable than pure PLA which can be observed in corrected Figure 6.
Reviewer #1:
Paragraph 2.2 The paragraph should be rewritten reporting the information about the Brabender model used for sample preparation and all the conditions (i.e. temperature, screw speed etc) should be given. The authors report that flat strip samples are produced with a thickness of 1 mm. I suppose that a flat slit extrusion head has been used. The film were then collected on a rotating spool, can the authors give more details about the spinning speed. Usually, compounding is performed in twin screw extruders and the pellets obtained therefrom are then processed in single screw extruders. The authors directly used a single screw, how were the polymers mixed to ensure the right mixing ratio. This is usually ensured by using gravimetric feeders. All the mechanical properties are reported with not evidence of error bars and no statistical analysis is performed to assess the p-value. This should be done to support the data analysis.
Answer 3 for Reviewer #1: We appreciate Reviewer’s suggestions. In our Institution we can use only the single-screw extruder, where all heating zones have the same temperature. We rewrite Paragraph 2.2 as following: “The selected PLA and PHB pellets were firstly dried at 343 K in a dryer for 24h. Secondly PLA/PHB blends were prepared by manual mixing of selected pellets in a laboratory beaker according to composition presented in Table 1. The prepared mixture was processed using a single-screw laboratory extruder (Zamak Mercator, Poland) equipped with 25 mm screw diameter and characterized by a L/D ratio of 24. Temperature of extrusion process amounted to 453 K, the screw rotation speed was about 40 rpm and pressure equaled 17 atm. The flat tape of blends was formed by special flat slit extrusion head. The estimated output of the process amounted to 8 kg/h. The formed flat tape was cooled in the air at room temperature. Finally the tape was cut into smaller samples, which were 150 mm long, 25 mm wide and 1 mm thick.”
We changed plots and we added error bars.
Reviewer 2 Report
Review of “Thermoplastic and biodegradable polymer blends from plant materials” by O. Olejnik, A. Masek, J. Zawadzillo
In this paper aliphatic polyester blends (PLA/PHB) blends with different compositions have been prepared by extrusion. The processing properties (melt-volume flow rate and viscosity) and mechanical properties (tensile test) and rheology of the blends have been studied. Based on mixing energy the authors find that PHB act as plasticizer of PLA and based on DSC analysis they assume that the polymers are miscible.
I consider that in order to be published in “Materials” this paper needs some major amendments:
#1. Mechanical properties: the interpretation of the results is merely descriptive. A more detailed interpretation of the results is necessary.
#2. Figure 2, DSC curves: It is not possible to distinguish correctly the DSC curves of different samples. The curves should be shifted vertically, keeping a distance between them. In addition, different colors and/or types of line should be used to distinguish curves corresponding to different samples.
#3. Point 3.2 Differential Scanning Calorimetry (DSC) analysis: some curves present multiple melting peaks that should be explained.
#4. I think that the title of the manuscript is too general for its content. It should be included the name of the specific polymers used for the blends.
#5. In my opinion, the last paragraph of the Introduction is too generic. I think that a paragraph should be included describing the work done with more specific details and addressing the goal of the paper.
Minor ammendements
#6 Line 182: According to International Association of Thermal Analysis, “thermogram” word can only be used for TGA curves. For DSC, “DSC runs” or ·”DSC curves” should be used. The use of the “thermogram” word applied to DSC curves, should be corrected.
#7 References section: Reference 18 it is not correct, the correct is as follows:
Feng F, Ye LStructure and Property of Polylactide/Polyamide Blends. J. Macromol. Sci. B.. 2010, 49: 1117-1127.
Author Response
Institute of Polymer and Dye Technology
Technical University of Lodz
90-924 Lodz, ul Stefanowskiego 12/16, Poland
Tel.: +48 42 631 32 23, Fax: +48 42 636 25 43
January 11, 2020
Materials
Dear Professor,
We are resubmitting our revised paper entitled Thermoplastic and biodegradable polymer blends from plant materials by, Olga Olejnik, Anna Masek and Jakub Zawadziłło with a request to reconsider it for publication in Materials.
We have carefully considered the Editor and Reviewers' comments. The manuscript was revised exactly according to these comments. The list of responses to the reviewers’ comments and corrections made in the manuscript is attached.
The manuscript has not been previously published, is not currently submitted for review to any other journal, and will not be submitted elsewhere before a decision is made by this journal.
For correspondence please use the following information:
corresponding author: Anna Masek
Institute of Polymer and Dye Technology
Technical University of Lodz
90-924 Lodz, ul Stefanowskiego 12/16, Poland
Tel.: +48 42 631 32 93
Fax: +48 42 636 25 43
e-mail: anna.masek@p.lodz.pl
Yours sincerely,
Ph. D., D.Sc. Anna Masek
Answers to reviewer #2 comments
Reviewer #2:
Comments and Suggestions for Authors
Review of “Thermoplastic and biodegradable polymer blends from plant materials” by O. Olejnik, A. Masek, J. Zawadzillo In this paper aliphatic polyester blends (PLA/PHB) blends with different compositions have been prepared by extrusion. The processing properties (melt-volume flow rate and viscosity) and mechanical properties (tensile test) and rheology of the blends have been studied. Based on mixing energy the authors find that PHB act as plasticizer of PLA and based on DSC analysis they assume that the polymers are miscible.
I consider that in order to be published in “Materials” this paper needs some major amendments:
#1. Mechanical properties: the interpretation of the results is merely descriptive. A more detailed interpretation of the results is necessary.
Answer 1 for Reviewer #2: We thank Reviewer for paying attention to this issue. We added new plots with error bars, where results of pure PLA and PHB were also presented and some missing information is placed in text as following: “According to Figure 8. addition of small amount of PHB resulted in improvement of me-chanical properties. The similar observations were noticed by Zhang et. al [15], where the enhancement were explained as finely dispersion of PHB crystals in PLA matrix, thus these crystals acted as filler. Too large amount of PHB crystals could not be well dispersed and causes deterioration of mechanical properties.”
#2. Figure 2, DSC curves: It is not possible to distinguish correctly the DSC curves of different samples. The curves should be shifted vertically, keeping a distance between them. In addition, different colors and/or types of line should be used to distinguish curves corresponding to different samples.
Answer 2 for Reviewer #2: We are thankful for this comment. This Figure has been changed into more clear with curves vertically shifted keeping a distance between them.
#3. Point 3.2 Differential Scanning Calorimetry (DSC) analysis: some curves present multiple melting peaks that should be explained.
Answer 3 for Reviewer #1: We are grateful for Reviewer’s comments. We decided to add missing information: “The DSC curves of selected PLA/PHB blends are presented in Figure 2. According to this figure, blends: PLA/PHB 10/50, PLA/PHB 20/50 and PLA/PHB 30/40 are more miscible, while PLA/PHB 40/30 and PLA/PHB 50/10 blends are characterised by two visible melting peaks. The first peak belongs to the one type of crystal melting probably belonging to PLA, while the second one is likely related to PHB crystals melting, which was also detected by Zhang et. al. [15].”
#4. I think that the title of the manuscript is too general for its content. It should be included the name of the specific polymers used for the blends.
Answer 4 for Reviewer #1: The proposition of the new title.: “Processability and mechanical properties of thermoplastic and biodegradable polylactide/polyhydroxybutyrate (PLA/PHB) blends from plant materials”
#5. In my opinion, the last paragraph of the Introduction is too generic. I think that a paragraph should be included describing the work done with more specific details and addressing the goal of the paper.
Answer 5 for Reviewer #1: We thank Reviewer for paying attention to this problem. We add some extra paragraph: “Taking into account need for creating more environmentally friendly materials with enhanced mechanical properties and improved processability, we are focused on designing eco-friendly blends using different mass ratio of biopolymer pellets and find the best formulation without incorporating any additional substances.”
Minor amendments
#6 Line 182: According to International Association of Thermal Analysis, “thermogram” word can only be used for TGA curves. For DSC, “DSC runs” or ·”DSC curves” should be used. The use of the “thermogram” word applied to DSC curves, should be corrected.
Answer 6 for Reviewer #1: This mistake has been corrected.
#7 References section: Reference 18 it is not correct, the correct is as follows:
Feng F, Ye LStructure and Property of Polylactide/Polyamide Blends. J. Macromol. Sci. B.. 2010, 49: 1117-1127
Answer 7 for Reviewer #1: This mistake has been corrected.
Reviewer 3 Report
Comments: Major Revise
This manuscript focused on developing bio-based biodegradable material with boosted processing performance for the packaging industry. By optimizing the mass ratio of polylactide (PLA) and polyhydroxybutyrate (PHB) using extrusion process, the PLA/PHB blend of 50/10 was found to be the most suitable product. The mechanical and processing properties including mixing energy, DSC, melt flow rate, dynamic mechanical analysis of different mixture ratios were well explored and compared. However, improvement is in need of the current manuscript, especially from an innovation point.
- There are numerous studies and concepts mentioning eco-friendly materials for the packaging industry, like paper, polypropylene. In this study, traditional PLA and PHB were chosen, what’s the advantages/innovation for mixing PLA/DHB?
- There was only one value for each sample present in every test, please provide the parallel sample numbers for each test.
- The melt-volume flow rate (Figure 6), viscosity (Figure 7), maximum tensile strength, and elongation at ultimate strength (Figure 8) of PHB (100%) and PLA (100%) should be provided.
- Regarding the biodegradable property of the two-component polymer blends, especially considering the eco-friendly requirement, what’s the degradable period of the blends?
- This blend is also expected to be safe for human health, thus basic biological safety evaluation like cell toxicity should be included.
Minor revision:
- The title of 2.1. and 2.3. are the same.
- Complete the conclusion part, line 305.
Author Response
Institute of Polymer and Dye Technology
Technical University of Lodz
90-924 Lodz, ul Stefanowskiego 12/16, Poland
Tel.: +48 42 631 32 23, Fax: +48 42 636 25 43
January 11, 2020
Materials
Dear Professor,
We are resubmitting our revised paper entitled Thermoplastic and biodegradable polymer blends from plant materials by, Olga Olejnik, Anna Masek and Jakub Zawadziłło with a request to reconsider it for publication in Materials.
We have carefully considered the Editor and Reviewers' comments. The manuscript was revised exactly according to these comments. The list of responses to the reviewers’ comments and corrections made in the manuscript is attached.
The manuscript has not been previously published, is not currently submitted for review to any other journal, and will not be submitted elsewhere before a decision is made by this journal.
For correspondence please use the following information:
corresponding author: Anna Masek
Institute of Polymer and Dye Technology
Technical University of Lodz
90-924 Lodz, ul Stefanowskiego 12/16, Poland
Tel.: +48 42 631 32 93
Fax: +48 42 636 25 43
e-mail: anna.masek@p.lodz.pl
Yours sincerely,
Ph. D., D.Sc. Anna Masek
Answers to reviewer #3 comments
Reviewer #3:
This manuscript focused on developing bio-based biodegradable material with boosted processing performance for the packaging industry. By optimizing the mass ratio of polylactide (PLA) and polyhydroxybutyrate (PHB) using extrusion process, the PLA/PHB blend of 50/10 was found to be the most suitable product. The mechanical and processing properties including mixing energy, DSC, melt flow rate, dynamic mechanical analysis of different mixture ratios were well explored and compared. However, improvement is in need of the current manuscript, especially from an innovation point.
- There are numerous studies and concepts mentioning eco-friendly materials for the packaging industry, like paper, polypropylene. In this study, traditional PLA and PHB were chosen, what’s the advantages/innovation for mixing PLA/DHB?
Answer 1 for Reviewer #3: We thank Reviewer for paying attention to this issue. We are focused on creating totally bio-based and biodegradable materials which could replace traditional thermoplastics, including polypropylene which is not biodegradable in environmental conditions. On the other hand, paper does not have required mechanical properties and also needs additional treatment, such as lamination which deteriorates biodegradation. Blending environmentally friendly polymer like PLA and PHB contributes to achieve novel materials with improved processing as well as mechanical properties than pure ones and let to preserve its biodegradability. Therefore such materials have suitable mechanical properties and are totally environmentally friendly. In this research we are mostly focused on processability and mechanical properties of created blends.
- There was only one value for each sample present in every test, please provide the parallel sample numbers for each test.
- The melt-volume flow rate (Figure 6), viscosity (Figure 7), maximum tensile strength, and elongation at ultimate strength (Figure 8) of PHB (100%) and PLA (100%) should be provided.
Answer 2 and 3 for Reviewer #3: We are grateful for Reviewer’s comments. We added new plots with error bars, where results for pure PLA and PHB were also presented. Only MFI results for pure PHB were impossible to obtain using our device because of too fast material flowing.
Answer 3 for Reviewer #3:
- Regarding the biodegradable property of the two-component polymer blends, especially considering the eco-friendly requirement, what’s the degradable period of the blends?
Answer 4 for Reviewer #3: We are thankful for this question. Manufacturers provide, that PLA and PHB pellets used in this research belong to the biodegradable materials. In the current research we were mainly focused on processability and mechanical properties of our blends and its biodegradability can be described in next article.
- This blend is also expected to be safe for human health, thus basic biological safety evaluation like cell toxicity should be included.
Answer 5 for Reviewer #3: We thank the Reviewer for paying attention to this issue. Manufacturers provide, that PLA and PHB pellets can be applied for use in fresh food packaging and food serviceware applications, nevertheless further analysis of its biological safety can be presented in next research. In the current research we were mainly focused on processability and mechanical properties
Minor revision:
- The title of 2.1. and 2.3. are the same.
Answer 6 for Reviewer #3: The mistake has been corrected. The title of 2.1 is named “Reagents”.
- Complete the conclusion part, line 305
Answer 7 for Reviewer #3: The conclusion has been completed. “Presented blend has the potential to replace non-biodegradable materials containing unsafe for human beings additives and become more popular than traditional thermoplastics”
Reviewer 4 Report
The novelty of this study is the preparation of two-component polymer blends based on PLA and PHB by mixing their different mass ratios without using of any additive by extrusion at 453 K. Blends were characterized by DSC, DMA and the melt flow rate, as well mixing energy, torque and mechanical properties (maximum tensile strength, tensile strength at break, elongation at break and elongation at maximum strength) were also measured.
However, the authors should carefully revise the manuscript, starting with the title and make it more correct. PHB is not plant material. In addition, the experiments for biodegradability of the prepared blends have not been performed. Both of polymers are biodegradable, but it is not shown what happens during thermal extrusion of these two immiscible polymers. So, I recommend modification of the TITLE in order to highlight the emphasis of the study.
The ABSTRACT is longer than 200 words. Please revise it and you need to shorten the background – the text from L14 to L19; please include brief description of the main methods and summarize the article's main findings.
Keywords should be change – bio-based and biodegradable are not specific.
INTRODUCTION must be significantly improved. The current state of the research field should be reviewed carefully and key publications cited. L32-L34 is refer only for PLA. Detailed comparison with the well-known about the miscibility/processing aspects of PLA/PHB blends and proposed in the manuscript should be done. In the present study it is not clear why the authors have chosen selected mass ratios. Moreover, Plavec et al. [Polymer Testing 92 (2020) 106880], Hu et al. [Polymer 2008, 49, 4204-4210], as well Zhang et al. [Adv. Polym. Technol. 2011, 30, 67–79] have already described in details such preparation of PLA/PHB blends. In this relation, please highlight controversial and diverging hypotheses. Concerning the PHB, a problem related to its processing is appeared because the small difference between the melting (180°C) and degradation temperatures (240°C). In addition, because of its high crystallinity, PHB is stiff and brittle, and this results in very poor mechanical properties with a low extension at break. For these reasons more efforts have been aimed at blending the PHB with appropriate agents which are able to reduce the crystallization, thus decreasing the brittleness and therefore improving the overall physical properties and processability of PHB. Therefore, the sentence in L72 – “PHB has different mechanical and thermal properties that makes it 72 a very attractive material for making polymer blends” is not correct.
The discussion is fails, especially in the explanation of thermal interactions between PHB and PLA, and mechanical properties as well.
Conclusions should be reconsider. Abbreviations are given yet.
For all of these reasons, I recommend the publication of the manuscript as full paper, but only after major and careful revision.
Additional comments:
SI Units are recommended to use. However, in the case of DSC (glass transition and melting temperatures) is more appropriate to give it in Celsius than in Kelvin. In this relation, check L122 – heating rate of 10 K/min.
L98 – what means “dried resins”? If the temperature is 343 K, the polymers have not melted yet.
Table 1 – weight or mass ratio?
L136-L137 – it is not clearly described how exactly from flat strip samples with a width of 25 mm and thickness of 1 mm, a flat strip test pieces (specimens) with width of 25 mm and length of 80 mm were prepared for mechanical testing.
L167 – “PHB acts like a plasticizer”. How exactly one stiff and brittle polymer can be plasticizer?
Figure 2 – the caption of Y axis is not correct. The DSC curves must be given one by one in different colors and the Tg and Tm for each curve should be mark clearly.
Section 3.2. should be reconsidered and write clearer. By DSC is not possible to detect the degradation of polymers. Pure PHB has sharp melting peak and no discernible glass transition. What about the recrystallization peaks and what they indicated? The data from Figures 3, 4, and 5 can be summarize in Table – this will be more readable.
Improve discussion in the Section 3.3. Please, provide stress–strain curves and include the values of modulus of elasticity.
Figure 10 should be provided with high quality resolution.
Figure 13 is redundant.
L288-L289 – Having in mind that pure PHB has very low extension at break, how the authors can explain the rises of elongation at break with increases the PHB content?
Author Response
Institute of Polymer and Dye Technology
Technical University of Lodz
90-924 Lodz, ul Stefanowskiego 12/16, Poland
Tel.: +48 42 631 32 23, Fax: +48 42 636 25 43
January 11, 2021
Materials
Dear Professor,
We are resubmitting our revised paper entitled Thermoplastic and biodegradable polymer blends from plant materials by, Olga Olejnik, Anna Masek and Jakub Zawadziłło with a request to reconsider it for publication in Materials.
We have carefully considered the Editor and Reviewers' comments. The manuscript was revised exactly according to these comments. The list of responses to the reviewers’ comments and corrections made in the manuscript is attached.
The manuscript has not been previously published, is not currently submitted for review to any other journal, and will not be submitted elsewhere before a decision is made by this journal.
For correspondence please use the following information:
corresponding author: Anna Masek
Institute of Polymer and Dye Technology
Technical University of Lodz
90-924 Lodz, ul Stefanowskiego 12/16, Poland
Tel.: +48 42 631 32 93
Fax: +48 42 636 25 43
e-mail: anna.masek@p.lodz.pl
Yours sincerely,
Ph. D., D.Sc. Anna Masek
Answers to Revewer#4 comments
Comments and Suggestions for Authors
The novelty of this study is the preparation of two-component polymer blends based on PLA and PHB by mixing their different mass ratios without using of any additive by extrusion at 453 K. Blends were characterized by DSC, DMA and the melt flow rate, as well mixing energy, torque and mechanical properties (maximum tensile strength, tensile strength at break, elongation at break and elongation at maximum strength) were also measured.
Reviewer#4: However, the authors should carefully revise the manuscript, starting with the title and make it more correct. PHB is not plant material. In addition, the experiments for biodegradability of the prepared blends have not been performed. Both of polymers are biodegradable, but it is not shown what happens during thermal extrusion of these two immiscible polymers. So, I recommend modification of the TITLE in order to highlight the emphasis of the study.
Answer 1 to Reviewer#4: We appreciate Reviewer’s suggestions and we changed the title of our article.
Reviewer#4: The ABSTRACT is longer than 200 words. Please revise it and you need to shorten the background – the text from L14 to L19; please include brief description of the main methods and summarize the article's main findings.
Answer 2 to Reviewer#4: We thank Reviewer for paying attention to this issue. We revised the abstract, shorten the background as following: “This work considered the application of ecofriendly biodegradable materials basing on polylactide and polyhydroxybutyrate instead of conventional polymeric materials, in order to prevent further environmental endangerment by accumulation of synthetic petro-materials. The new approach to this topic is focused on analysing processing properties of PLA/PHB blends without incorporating any additive, which could have a harmful impact on human organisms, including the endocrine system. The main aim of the research was to find the best PLA/PHB ratio to obtain material with desirable mechanical, processing as well as application properties. Therefore, two-component polymer blends were prepared by mixing different mass ratios of polylactide (PLA) and polyhydroxybutyrate (PHB) (100/0, 50/10, 50/20, 40/30, 50/50, 30/40, 20/50, 10/50 and 0/100 mass ratio) using extrusion process. The prepared blends were analyzed also in terms of thermal and mechanical properties, as well as miscibility and surface characteristics. Taking into account the test results, the PLA/PHB blend of 50/10 ratio turned out to be the most suitable in terms of mechanical and processing properties. This blend has the potential to become a bio-based and simultaneously biodegradable material safe for human health dedicated to packaging industry.”
Reviewer#4: Keywords should be change – bio-based and biodegradable are not specific.
Answer 3 to Reviewer#4: We are grateful for Reviewer’s comment and we removed keywords, including “bio-based” and “biodegradable”, which are not specific. We added keyword: bioplastic.
INTRODUCTION must be significantly improved. The current state of the research field should be reviewed carefully and key publications cited. L32-L34 is refer only for PLA. Detailed comparison with the well-known about the miscibility/processing aspects of PLA/PHB blends and proposed in the manuscript should be done. In the present study it is not clear why the authors have chosen selected mass ratios. Moreover, Plavec et al. [Polymer Testing 92 (2020) 106880], Hu et al. [Polymer 2008, 49, 4204-4210], as well Zhang et al. [Adv. Polym. Technol. 2011, 30, 67–79] have already described in details such preparation of PLA/PHB blends. In this relation, please highlight controversial and diverging hypotheses. Concerning the PHB, a problem related to its processing is appeared because the small difference between the melting (180°C) and degradation temperatures (240°C). In addition, because of its high crystallinity, PHB is stiff and brittle, and this results in very poor mechanical properties with a low extension at break. For these reasons more efforts have been aimed at blending the PHB with appropriate agents which are able to reduce the crystallization, thus decreasing the brittleness and therefore improving the overall physical properties and processability of PHB. Therefore, the sentence in L72 – “PHB has different mechanical and thermal properties that makes it 72 a very attractive material for making polymer blends” is not correct.
The discussion is fails, especially in the explanation of thermal interactions between PHB and PLA, and mechanical properties as well.
Conclusions should be reconsider. Abbreviations are given yet.
For all of these reasons, I recommend the publication of the manuscript as full paper, but only after major and careful revision.
Answer 4 to Reviewer#4: We thank Reviewer for paying attention to this issue. We removed incorrect sentence “PHB has different mechanical and thermal properties that makes it 72 a very attractive material for making polymer blends” and some information about PLA and PHB. We focused on PLA/PHB blends and added information about them:
“Both polylactide (PLA) and polyhydroxybutyrate (PHB) are bio-based as well as bio-degradable polyesters and are characterized by good biocompatibility, biodegradability and sustainability. These polymers can be compared with conventionally used plastics by the thermal and mechanical properties [22]. Therefore, Polyhydroxybutyrate and polylactide can be used as transparent films in the food packaging industry because of their bio-degradability and biocompatibility that is extremely necessary nowadays [23]. PLA and PHB are also suitable for production of disposable cutlery [24] and biomedical implants [25]. Separately PLA and PHB are characterized by poor processing [26] but such proper-ties of their blends were not interpreted. Moreover, these polymers are brittle at room temperature. Nevertheless, blending polylactide with polyhydroxybutyrate which is a highly crystalline biopolyester, result in obtaining new materials with interesting physical, thermal and mechanical properties in comparison to neat polylactide (PLA) [27]. Blending polylactide with polyhydroxybutyrate is not a new idea [28,29]. Based on the information available in the literature, mixing PLA with 25 wt% of PHB provided reinforcement effect and better mechanical properties in comparison to neat PLA., which was observed by Zhang et. al. [15]. Furthermore, some interactions between both polymers were also noticed. Crystallization behavior of poly(L-lactic acid) affected by the addition of a small amount of poly(3-hydroxybutyrate) was researched by Hu et.al. [30]. The impact of different molecular weight PHB on PLA matrix was also detected. Nevertheless, most of such researches concerned poly(3-hydroxybutyrate) and there is lack of information about commercially available PHB containing 12 mol% 4-hydroxybutyrate, which could be interesting not only for scientist but also for producers.
Most of researches concerning PLA/PHB blends are mainly focused on testing films and there is no information about thicker forms of specimens which could be dedicated for stiffer packaging, including boxes. The processability of such blends are also insufficiently described. Furthermore, the best mass ratio of PLA and PHB to obtain blend with the most satisfactory properties was not reported. Thus, we decided to create PLA/PHB blends from commercially available pellets to obtain material with desirable mechanical, processing as well as application properties dedicated to environmentally friendly pack-aging.”
Additional comments:
SI Units are recommended to use. However, in the case of DSC (glass transition and melting temperatures) is more appropriate to give it in Celsius than in Kelvin. In this relation, check L122 – heating rate of 10 K/min.
Answer to Reviewer#4: We thank Reviewer for the suggestion. Nevertheless, we would like to use the same temperature units in the whole paper to maintain order. Heating rate amounted to generally 10 temperature degrees/min and in our opinion can be presented as 10°C/min or 10 K/min, depending on which unit we are operating
Reviewer#4: L98 – what means “dried resins”? If the temperature is 343 K, the polymers have not melted yet.
Answer to Reviewer#4: We thank the Reviewer for paying attention to this issue. “Dried resins” are resins/pellets after drying in the oven to eliminate possible absorbed water on the surface of the particles. Such granulates can be processed, unlike to granules before drying, which contain humidity undesirable during processing.
Reviewer#4: Table 1 – weight or mass ratio?
Answer to Reviewer#4: We thank Reviewer for paying attention to this inadvertence. We changed “weight ratio” into “mass ratio” in Table 1.
Reviewer#4: L136-L137 – it is not clearly described how exactly from flat strip samples with a width of 25 mm and thickness of 1 mm, a flat strip test pieces (specimens) with width of 25 mm and length of 80 mm were prepared for mechanical testing.
Answer to Reviewer#4: We are grateful for Reviewer’s comment. We improved description of preparation as follow: “The selected PLA and PHB pellets were firstly dried at 343 K in a dryer for 24h. Then, PLA/PHB mixtures were prepared by manually mixing of selected pellets in a laboratory beaker according to composition presented in Table 1. The prepared mixture was pro-cessed using a single-screw laboratory extruder (Zamak Mercator, Poland) equipped with 25 mm screw diameter and characterized by a L/D ratio of 24. Temperature of extrusion process amounted to 453 K, the screw rotation speed was about 40 rpm and pressure equaled 17 atm. The flat tape of blends was formed by special flat slit extrusion head. The estimated output of the process amounted to 8 kg/h. The formed flat tape was cooled in the air at room temperature. Finally the tape was cut into smaller samples, which were 150 mm long, 25 mm wide and 1 mm thick.”
Reviewer#4: L167 – “PHB acts like a plasticizer”. How exactly one stiff and brittle polymer can be plasticizer?
Answer to Reviewer#4: We are thankful for Reviewer’s comment. Most of all, we wanted to show, that addition of PHB to PLA led to facilitate the processing of PLA, which has been revealed in mixing energy results. The higher the amount of PHB, the lower the mixing energy is. Moreover, mixing PLA with PHB causes decreasing in Tg of the blend.
Reviewer#4: Figure 2 – the caption of Y axis is not correct. The DSC curves must be given one by one in different colors and the Tg and Tm for each curve should be mark clearly.
Answer to Reviewer#4: We are thankful for this comment. This Figure has been changed into more clear with curves vertically shifted keeping a distance between them.
Figure 2. DSC curves of polylactide (PLA), polyhydroxybutyrate (PHB) and their blends.
Reviewer#4: Section 3.2. should be reconsidered and write clearer. By DSC is not possible to detect the degradation of polymers. Pure PHB has sharp melting peak and no discernible glass transition. What about the recrystallization peaks and what they indicated? The data from: Figures 3, 4, and 5 can be summarize in Table – this will be more readable.
Answer to Reviewer#4: We thank Reviewer for this comment and we summarized Figures 3, 4, and 5 in Table to provide better readability.
Table 2. Composition of two-component polymer blends based on polylactide(PLA) and polyhydroxybutyrate (PHB).
|
Sample |
PHB content [%] |
Tg [K] |
ΔHcc [J/g] |
Tcc [K] |
ΔHm [J/g] |
Tm [K] |
|
PLA |
0 |
335 |
14,6 |
400 |
15.6 |
429 |
|
PHB |
100 |
317 |
15.4 |
355 |
33.9 |
439 |
|
PLA/PHB 50/10 |
17 |
332 |
23.1 |
394 |
19.8 |
426 |
|
PLA/PHB 40/30 |
43 |
334 |
22.4 |
385 |
23.2 |
435 |
|
PLA/PHB 50/50 |
50 |
331 |
22.4 |
379 |
26.0 |
436 |
|
PLA/PHB 30/40 |
57 |
329 |
19.5 |
374 |
24.8 |
438 |
|
PLA/PHB 20/50 |
71 |
327 |
18.4 |
370 |
28.8 |
438 |
|
PLA/PHB 10/50 |
83 |
325 |
17.6 |
366 |
30.4 |
438 |
Differential scanning calorimetry provides information about temperatures of glass transition, cold crystallization and melting point. Moreover, this measurement can be useful for determining compatibility of polymer blends. It can be observed that phase transition temperature of composites depends on the intermolecular interactions between the chains of these polymers. In the case of compatible materials, the interaction forces are stronger and a single temperature for a specific phase transition is seen. If a composite consists of incompatible components, two temperatures for glass transition, crystallization and melting point are visible in the DSC curve. The DSC curves of selected PLA/PHB blends are presented in Figure 2. According to this figure, PLA/PHB blends are characterised by two visible melting peaks. The first peak belongs to the one type of crystal melting probably belonging to PLA, while the second one is likely related to PHB crystals melting, which was also detected by Zhang et. al. [15]. Nevertheless, the second peak of every blend is responsible for whole material melting and this parameter is noted in Table 2 as melting point (Tm).
According to Figure 2 and Table 2., blends with equal or higher polyhydroxybutyrate (PHB) content have lower glass transition (Tg) temperatures than materials with higher amount of polylactide (PLA). This parameter decreases for these blends in a linear way from 333 K to 325 K as a function of polyhydroxybutyrate (PHB) content. The addition of smaller amounts of polyhydroxybutyrate (PHB) to polylactide (PLA) causes a drop of this parameter but in an irregular way. Nevertheless, it can be assumed that the polyhydroxybutyrate (PHB) acts like a plasticizer in relation to polylactide, because every blend containing this biopolyester has lower Tg in comparison to pure polylactide. Furthermore, only one glass transition is observed in the DSC.
According to Figure 2 and Table 2, the temperature of cold crystallization (Tcc), similarly to glass transition temperature (Tg), decreases with increase of polyhydroxybutyrate content. This parameter changes in a linear way. The blend containing the lowest amount of polyhydroxybutyrate (PHB) has the highest value of about 394 K which is lower than for pure PLA by about 6 degrees. The blend with the highest PHB content has the lowest cold crystallization temperature of about 366 K, which is higher than the value for pure PHB by about 13 degrees. It means that the higher PHB content is, the faster blend crystalizes in lower temperatures. The enthalpies of blends phase changes also shows, that addition of PHB leads to increase in crystal phase, which was observed also by Hu et.al. [30].
Blending a small amount of polyhydroxybutyrate (PHB) with polylactide (PLA) causes a decrease in melting point from 429 K to 426 K in comparison to pure PLA which can be noticed in Figure 3 (c). Polymer blends of 43% and higher PHB content reveal higher melting point than pure polylactide. It can be seen, that only a small addition of polyhydroxybutyrate can be useful for reducing the melting point which corresponds to the material processing temperature and acts as a plasticizer.
Reviewer#4: Improve discussion in the Section 3.3. Please, provide stress–strain curves and include the values of modulus of elasticity.
Answer to Reviewer#4: We thank the Reviewer for paying attention to this issue. Unfortunately we have only pdf versions of stress-strain curves which can not be edited and added to following paper. Also modulus of elasticity can not be added at the moment.
Reviewer#4: Figure 10 should be provided with high quality resolution.
Answer to Reviewer#4:We provided Figure 10 with high quality resolution.
Reviewer#4: Figure 13 is redundant.
Answer to Reviewer#4: We removed Figure 13.
Reviewer#4: L288-L289 – Having in mind that pure PHB has very low extension at break, how the authors can explain the rises of elongation at break with increases the PHB content?
Answer to Reviewer#4: We appreciate Reviewer’s suggestions. We assumed, that interactions between PLA and PHB in a blend can cause this phenomenon.
Round 2
Reviewer 1 Report
The paper was resubmit but no substantial improvement was made that can justify the acceptance in the present form
Author Response
Institute of Polymer and Dye Technology
Technical University of Lodz
90-924 Lodz, ul Stefanowskiego 12/16, Poland
Tel.: +48 42 631 32 23, Fax: +48 42 636 25 43
January 31, 2021
Materials
Dear Professor,
We are resubmitting our revised paper entitled Processability and mechanical properties of thermoplastic polylactide/polyhydroxybutyrate (PLA/PHB) bioblends by, Olga Olejnik, Anna Masek and Jakub Zawadziłło with a request to reconsider it for publication in Materials.
We have carefully considered the Editor and Reviewers' comments. The manuscript was revised exactly according to these comments. The list of responses to the reviewers’ comments and corrections made in the manuscript is attached.
The manuscript has not been previously published, is not currently submitted for review to any other journal, and will not be submitted elsewhere before a decision is made by this journal.
For correspondence please use the following information:
corresponding author: Anna Masek
Institute of Polymer and Dye Technology
Technical University of Lodz
90-924 Lodz, ul Stefanowskiego 12/16, Poland
Tel.: +48 42 631 32 93
Fax: +48 42 636 25 43
e-mail: anna.masek@p.lodz.pl
Yours sincerely,
Ph. D., D.Sc. Anna Masek
Answer to reviewer #1 comments
Comments and Suggestions for Authors:
The paper was resubmit but no substantial improvement was made that can justify the acceptance in the present form
Answer: We are thankful for the Reviewer’s suggestions and we attempted to meet the all expectations. Nevertheless, most of comments are too general and we do not know how to refer to them and how to make better improvements in our manuscript. We used all information available and described it at its best form. Moreover, the further language correction has been already performed.
Reviewer 2 Report
I find that the authors have correctly adressed the comment I proposed. In my opinion, the paper has been greately improved and it could be published.
Author Response
Institute of Polymer and Dye Technology
Technical University of Lodz
90-924 Lodz, ul Stefanowskiego 12/16, Poland
Tel.: +48 42 631 32 23, Fax: +48 42 636 25 43
January 31, 2021
Materials
Dear Professor,
We are resubmitting our revised paper entitled Processability and mechanical properties of thermoplastic polylactide/polyhydroxybutyrate (PLA/PHB) bioblends by, Olga Olejnik, Anna Masek and Jakub Zawadziłło with a request to reconsider it for publication in Materials.
We have carefully considered the Editor and Reviewers' comments. The manuscript was revised exactly according to these comments. The list of responses to the reviewers’ comments and corrections made in the manuscript is attached.
The manuscript has not been previously published, is not currently submitted for review to any other journal, and will not be submitted elsewhere before a decision is made by this journal.
For correspondence please use the following information:
corresponding author: Anna Masek
Institute of Polymer and Dye Technology
Technical University of Lodz
90-924 Lodz, ul Stefanowskiego 12/16, Poland
Tel.: +48 42 631 32 93
Fax: +48 42 636 25 43
e-mail: anna.masek@p.lodz.pl
Yours sincerely,
Ph. D., D.Sc. Anna Masek
Answer to reviewer #2 comments
Comments and Suggestions for Authors:
I find that the authors have correctly adressed the comment I proposed. In my opinion, the paper has been greately improved and it could be published..
Answer: We are thankful for the Reviewer’s comments. Moreover, the further language correction has been already made.
Reviewer 3 Report
Comments: Accept after Minor Revision
This manuscript was improved after the revision. It can be accepted after minor revision. Written and language still need to be improved.
Author Response
Institute of Polymer and Dye Technology
Technical University of Lodz
90-924 Lodz, ul Stefanowskiego 12/16, Poland
Tel.: +48 42 631 32 23, Fax: +48 42 636 25 43
January 31, 2021
Materials
Dear Professor,
We are resubmitting our revised paper entitled Processability and mechanical properties of thermoplastic polylactide/polyhydroxybutyrate (PLA/PHB) bioblends by Olga Olejnik, Anna Masek and Jakub Zawadziłło with a request to reconsider it for publication in Materials.
We have carefully considered the Editor and Reviewers' comments. The manuscript was revised exactly according to these comments. The list of responses to the reviewers’ comments and corrections made in the manuscript is attached.
The manuscript has not been previously published, is not currently submitted for review to any other journal, and will not be submitted elsewhere before a decision is made by this journal.
For correspondence please use the following information:
corresponding author: Anna Masek
Institute of Polymer and Dye Technology
Technical University of Lodz
90-924 Lodz, ul Stefanowskiego 12/16, Poland
Tel.: +48 42 631 32 93
Fax: +48 42 636 25 43
e-mail: anna.masek@p.lodz.pl
Yours sincerely,
Ph. D., D.Sc. Anna Masek
Answer to reviewer #3 comments
Comments and Suggestions for Authors:
This manuscript was improved after the revision. It can be accepted after minor revision. Written and language still need to be improved.
Answer: We are thankful for the Reviewer’s comments. The further language correction has been already made. To make better improvements in our manuscript, we need exact suggestion what to change.
Reviewer 4 Report
The authors have taken into account my remarks and comments.
Nevertheless, some minor modifications are still required:
L13 - include the abbreviations of PHB and PLA here instead of L19/20
L125 - in point 2.4. should be mentioned is the heat-cool-heat cycles are used, or only heat-cool.
In my opinion the revised manuscript has been significantly improved and can be published in Materials when the aforementioned remarks are made.
Author Response
Institute of Polymer and Dye Technology
Technical University of Lodz
90-924 Lodz, ul Stefanowskiego 12/16, Poland
Tel.: +48 42 631 32 23, Fax: +48 42 636 25 43
January 31, 2021
Materials
Dear Professor,
We are resubmitting our revised paper entitled Processability and mechanical properties of thermoplastic polylactide/polyhydroxybutyrate (PLA/PHB) bioblends by, Olga Olejnik, Anna Masek and Jakub Zawadziłło with a request to reconsider it for publication in Materials.
We have carefully considered the Editor and Reviewers' comments. The manuscript was revised exactly according to these comments. The list of responses to the reviewers’ comments and corrections made in the manuscript is attached.
The manuscript has not been previously published, is not currently submitted for review to any other journal, and will not be submitted elsewhere before a decision is made by this journal.
For correspondence please use the following information:
corresponding author: Anna Masek
Institute of Polymer and Dye Technology
Technical University of Lodz
90-924 Lodz, ul Stefanowskiego 12/16, Poland
Tel.: +48 42 631 32 93
Fax: +48 42 636 25 43
e-mail: anna.masek@p.lodz.pl
Yours sincerely,
Ph. D., D.Sc. Anna Masek
Answer to reviewer #4 comments
Comments and Suggestions for Authors:
The authors have taken into account my remarks and comments.
Nevertheless, some minor modifications are still required:
L13 - include the abbreviations of PHB and PLA here instead of L19/20
L125 - in point 2.4. should be mentioned is the heat-cool-heat cycles are used, or only heat-cool.
In my opinion the revised manuscript has been significantly improved and can be published in Materials when the aforementioned remarks are made.
Answer: We are thankful for the Reviewer’s suggestions. We changed abbreviations in the abstract as following: This work considers the application of ecofriendly biodegradable materials based on polylactide (PLA) and polyhydroxybutyrate (PHB) instead of conventional polymeric materials, in order to prevent further environmental endangerment by accumulation of synthetic petro-materials. The new approach to this topic is focused on analyzing processing properties of blends without in-corporating any additive, which could have a harmful impact on human organisms, including the endocrine system. The main aim of the research was to find the best PLA/PHB ratio to obtain material with desirable mechanical, processing as well as application properties. Therefore, two-component polymer blends were prepared by mixing different mass ratios of PLA and PHB (100/0, 50/10, 50/20, 40/30, 50/50, 30/40, 20/50, 10/50 and 0/100 mass ratio) using extrusion process. The prepared blends were analyzed also in terms of thermal and mechanical properties, as well as miscibility and surface characteristics. Taking into account the test results, the PLA/PHB blend of 50/10 ratio turned out to be the most suitable in terms of mechanical and processing properties. This blend has the potential to become a bio-based and simultaneously biodegradable material safe for human health dedicated for packaging industry.
We also thoroughly described cycles of DSC measurement: “The analysis was conducted with the following parameters: First cycle (heating from 273 K to 473 K) under a dynamic flow of argon at a rate of 50 ml/min for 15 min, second cycle at 473 K under a dynamic flow of argon at a rate of 50 ml/min for 10 min. Third cycle (cooling from 473 K to 273 K) was performed under a dynamic flow of argon at a rate of 50 ml/min for 15 min., fourth cycle (heating from 273 K to 623 K under a dynamic flow of air at a rate of 50 ml/min The heating rate amounted to 20K/min. The temperatures of phase transitions of the samples and their specific heat were determined for the different compo-sitions on the basis of the DSC curves.”